# MetaUAS: Universal Anomaly Segmentation with One-Prompt Meta-Learning

**Bin-Bin Gao**
Tencent YouTu Lab, Shenzhen, China
csgaobb@gmail.com
Code and Models: https://github.com/gaobb/MetaUAS

## Abstract

Zero- and few-shot visual anomaly segmentation relies on powerful vision-language models that detect unseen anomalies using manually designed textual prompts. However, visual representations are inherently independent of language. In this paper, we explore the potential of a pure visual foundation model as an alternative to widely used vision-language models for universal visual anomaly segmentation. We present a novel paradigm that unifies anomaly segmentation into change segmentation. This paradigm enables us to leverage large-scale synthetic image pairs, featuring object-level and local region changes, derived from existing image datasets, which are independent of target anomaly datasets. We propose a one-prompt Meta-learning framework for Universal Anomaly Segmentation (MetaUAS) that is trained on this synthetic dataset and then generalizes well to segment any novel or unseen visual anomalies in the real world. To handle geometrical variations between prompt and query images, we propose a soft feature alignment module that bridges paired-image change perception and single-image semantic segmentation. This is the first work to achieve universal anomaly segmentation using a pure vision model without relying on special anomaly detection datasets and pre-trained visual-language models. Our method effectively and efficiently segments any anomalies with only one normal image prompt and enjoys training-free without guidance from language. Our MetaUAS significantly outperforms previous zero-shot, few-shot, and even full-shot anomaly segmentation methods.

## 1 Introduction

Visual anomaly classification (AC) and segmentation (AS) aims to group images and pixels into two different semantics, normal and anomalous, facilitating many applications such as industrial defect inspection in manufacturing [4, 39, 3, 74], medical image diagnosis [28, 65], and video surveillance [55, 40, 18], etc. Generally, AS is performed first, and then AC is obtained based on post-processing of the segmentation results. Therefore, AS is more essential than AC. AS can be viewed as binary semantic segmentation assuming that pixel-level annotated images are available. Unfortunately, it is usually difficult to collect anomaly images due to their scarcity in practical applications, and pixel-level annotations also involve labor costs. Based on available training images, e.g., only normal, normal with noise, few-shot normal or anomaly, and multi-class normal, have led to different types of AS tasks, such as unsupervised [47, 31, 72, 12, 37, 29], fully-unsupervised [9, 27, 38], few-shot [24, 14, 66, 69, 15], and unified AS [70, 17], etc. *These methods achieve excellent performance on seen objects but often perform poorly on unseen objects.*

To address this fragmentation, recent works, such as WinCLIP [26] and AnomalyCLIP [76], have attempted to design universal models, that are capable of recognizing anomalies for unseen objects. They typically build on vision-language models (i.e., CLIP [42]), benefiting from strong generalization

38th Conference on Neural Information Processing Systems (NeurIPS 2024).

ability. However, WinCLIP [26] still struggles with handcrafted text prompts about defects. Further, AnomalyCLIP [76] learns general prompt embedding through developing specialized modules but requiring fine-tuning on an auxiliary domain dataset with pixel-level annotations. In addition to being flexible, these vision-language-based methods are still weak in anomaly segmentation. Accurate anomaly segmentation is crucial in real-world applications, such as industrial inspection, as anomalies often relate to the area, shape, and location where they occur. On the other hand, we know that visual representations are not dependent on language in the animal world [16]. In particular, the principles of visual perception in non-human primates are very similar to humans [58]. Although there is room for universal AS based on visual-language models and worthwhile further to pursue, in this paper, *we want to explore how far we can go with a pure visual model without any guidance from language.*

To explore a more general AS framework, lets first review how the visual system perceives anomalies. Generally, humans can perceive anomalies when an input significantly deviates from those normal patterns stored in our brains. There is evidence to support this point in neuroscience. For example, predictive coding theory [43] postulates that the brain constantly generates and updates a "mental model". The mental model compares its expectations (or predictions) with the actual inputs from the visual cortex. This process allows the brain to perceive anomalies. In fact, PatchCore [47] captures normal local patch features, stores them in a memory bank, and recognizes anomalies by comparing input features with the memory bank. In addition, some distribution-based methods [12] learn a multivariate Gaussian distribution from normal local features and then utilize a distance metric to measure anomalies. However, these memory- and distribution-based methods usually require a certain number of normal images and thus are limited in universal (i.e., open-world) scenarios.

Actually, we can build similar concepts in AS. First, given one normal image prompt for each class, we take it as the expected output. Then, the actual input could be any query images from the same class of the normal prompt. Last but not least, how to construct a "mental model" to compare between a given normal image prompt and any query images. Despite these challenges, we can imagine that the "mental model" should satisfy several basic principles. First, it should have a strong generalization ability to perceive anomalies facing unseen objects or textures. Second, it can perform pixel-level anomaly segmentation only given one normal image prompt. Third, its training does not depend on target domain distribution or any guidance from language.

To obtain the "metal model", we rethink AS tasks and find they can be transformed into change segmentation between one normal image prompt and query images. From this novel perspective, we are capable of leveraging a larger number of synthetic image pairs that exhibit appearance changes based on available image datasets. We assume these synthesized image pairs carry mask annotations indicating change regions. Inspired by the "mental model" in predictive coding theory [43], we propose a simple but effective framework that learns the "metal model" in a one-prompt meta-learning manner. The meta-learning ensures strong generalization [8] when applying the model for segment unseen anomalies. Our contributions are summarized as follows:

- We present a novel paradigm that unifies anomaly segmentation into change segmentation. This paradigm enables us to leverage large-scale synthetic image pairs with object-level and local region changes, thereby overcoming the long-standing challenge of lacking large-scale anomaly segmentation datasets.
- We propose a one-prompt meta-learning framework training on synthesized images and generalizing well on real-world scenarios. To handle geometrical variations between prompt and query images, we proposed a soft feature alignment module that builds a bridge between paired-image change perception and singe-image semantic segmentation.
- We provide a pure visual foundation model for universal anomaly segmentation that can serve as an alternative to widely used vision-language models. Our method, which requires only a single normal image prompt and no additional training, effectively and efficiently segments any visual anomalies. On three industrial anomaly benchmarks, our approach achieves state-of-the-art performance, while also enjoying faster speed and requiring fewer parameters.

## 2   Related Work

**Unsupervised AS** aims to segment anomaly pixels for both normal and anomaly testing images only given full normal training images. Unsupervised AS can be categorized into two learning paradigms, separated and unified models. Most AS methods focus on training separated models for different

objects or textures. However, this separated paradigm may be impractical, as it requires high memory consumption and storage burden, especially with the number of classes increasing. In contrast, the unified models attempt to detect anomalies for all categories using a single model. Compared to the separated mode, the unified paradigm is more challenging as it requires handling more complex data distributions.

From a modeling perspective, AS methods can be roughly grouped into three groups, embedding, discriminator, and reconstruction. Embedding-based methods, such as PaDiM [12], MDND [44], PatchCore [47], CS-Flow [49] and PyramidFlow [29], assume that offline features extracted from a pre-trained model preserve discriminative information and thus help to separate anomalies from normal samples. They usually model normal features to a normal distribution or store them in a memory bank. Then, anomaly scores are calculated by comparing testing features and the modeled distribution or the memory bank. Discriminator-based methods, such as CutPaste [31], DRAEM [72], [10], and SimpleNet [37], typically convert unsupervised AS to supervised ones by introducing pseudo (synthesized) anomaly samples. The pseudo-anomaly samples are generated by pasting random patches or adding Gaussian noise to normal images or features. Naturally, a binary anomaly classifier or segmentation model can be trained on normal and pseudo-anomaly samples in a supervised manner. Reconstruction-based AS, such as autoencoder [59, 2, 20, 23], generative adversarial networks [41, 67, 71] and reconstruction networks [73, 45, 36], assume that anomalous regions should not be able to be properly reconstructed and thus result in high reconstruction errors since they do not exist in normal training samples. These methods tend to be computationally expensive because they involve reconstruction in image space. The recent knowledge distillation [5, 62, 61, 52, 13] or feature reconstruction methods [70, 75, 68] train a student or reconstruction network to match a fixed pre-trained teacher network and achieve a good balance between effectiveness and efficiency. However, all these methods are limited to recognizing anomalies in a close set as the same as the training set but often perform poorly on unseen classes in open-world scenarios.

**Few-shot AS** pays attention to learning with only a limited number of normal samples. TDG [54] proposes a multi-scale hierarchical generative model, which jointly learns self-supervised discriminator and generator in an adversarial training manner. DifferNet [48] detects defects utilizing a normalizing-flow-based density estimation from a few normal image features. RegAD [24] learns the category-agnostic feature registration, enabling the model to detect anomalies in new categories given a few normal images without fine-tuning. GraphCore [66] utilizes graph representation and provides a visual isometric invariant feature. FastRecon [15] utilizes a few normal samples as a reference to reconstruct a normal version for a query sample with distribution regularization, where the final anomaly detection can be achieved by sample alignment. Some works [14, 69] consider another few-shot setting where a limited number of samples is given from the anomalous classes. Instead of learning few-shot models with a few normal or anomaly images, we push it to a new extreme only using one normal image as a visual prompt at the inference stage, not involving model training.

**Zero- and Few-shot AS** mainly utilizes large pre-trained vision-language models, e.g., CLIP [42], have shown unprecedented generality, and achieved impressive performance. WinCLIP [26] firstly utilizes multiple handcrafted textual prompts on a powerful CLIP model that can yield excellent zero- and few-shot AS performance. AnomalyCLIP [76] learns object-agnostic text prompts that capture generic normality and abnormality in an image regardless of its foreground objects. InCtrl [77] detects residual CLIP visual features between test images and in-context few-shot normal samples. However, optimizing AnomalyCLIP [76] and InCtrl [77] requires an auxiliary domain dataset including normal and anomaly images. PromptAD [32] introduces the concept of explicit anomaly margin, which mitigates the training challenge caused by the absence of anomaly images. Furthermore, AnomalyGPT [21] incorporates a visual-language model and a large language model applying multi-turn dialogues. It not only indicates the presence and location of the anomaly but also provides a detailed description of anomalies. Instead of visual-language models, ACR [30] and MuSc [33] perform zero-shot AS only requiring information from batch- and full-level testing images, but they may be limited in privacy protection scenarios. Overall, existing most methods primarily use textual prompts based on visual-language models to identify anomalies. Different from these methods, we explore universal AS using one normal image as a visual prompt without guidance from language or information from testing images.

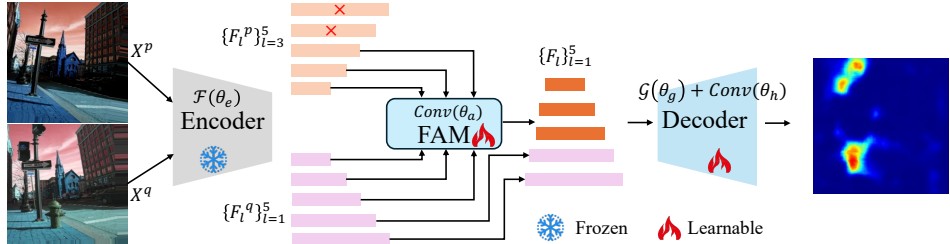

Figure 1: The proposed MetaUAS consists of an encoder, a feature alignment module (FAM), and a decoder. It is trained on a synthesized dataset in a one-prompt meta-learning manner for change segmentation tasks. Once trained, it can segment any anomalies providing only one normal image prompt.

# 3 Method

## 3.1 Rethinking Anomaly Segmentation

Unlike traditional image segmentation, since anomaly appearance has various ways, it is hard to exhaustively pre-define and collect enough anomaly samples to train an AS model. Most unsupervised AS methods have to use normal samples for building models. However, these unsupervised models are limited to recognizing anomalies in a close set but often perform poorly on unseen classes in open-world scenarios. In contrast, humans can quickly learn novel concepts from a few training examples. To this end, few-shot AS aims to adapt novel classes by only providing a few normal images. Unfortunately, existing few-shot AS models are still far behind unsupervised ones.

Zero- and few-shot AS methods rely on powerful vision-language models that are capable of handling unseen anomalies, benefiting from their strong generalization. For further boosting zero-shot performance, some recent works attempt to optimize models by an auxiliary domain dataset including normal and anomaly images with pixel-level annotations. Although they can generalize to different domains, training models with normal and anomaly images conflict with the original intentions of anomaly segmentation to some extent. In addition, visual representations are not dependent on language prompts. Given this perspective, a natural question emerges: *can an image prompt visual model replace textual prompts vision-language approaches for universal anomaly segmentation? And can such a one-prompt vision model be trained on a non-anomaly segmentation dataset?*

This paper explores and attempts to answer the above questions. Generally, anomalies mainly include "appearance", "disappearance", and "exchange", which are very similar to the types of changes in change segmentation [60]. Change segmentation aims to identify changes that occur on a pair of images captured at different times. Therefore, anomaly segmentation will be absorbed into change segmentation if we regard normal prompt and query as a pair of images captured at different times. Indeed, one can imagine that if a model is capable of perceiving changes, it would naturally generalize to anomaly segmentation. This simple transformation allows us to achieve universal AS with visual modality alone. This reason is change segmentation does not require anomaly images, as they can be trained using pairs of images containing any changes, which are easily synthesized by available image datasets.

## 3.2 One-Prompt Meta-Learning for Universal Anomaly Segmentation

Given a change segmentation dataset $D_{base} = \{X_i^p, X_i^q, Y_i\}_i$, where $(X_i^p, X_i^q)$ denotes the $i$-th image pair and $Y_i$ is its corresponding change mask. MetaUAS trains a meta-model based on the base set $D_{base}$ and then segments query images $\{X_i^q\}_{i=1}^N$ with the corresponding normal image prompt $X_i^p$ from a novel set $D_{novel}$, where $X_i^p$ and $X_i^q$ belongs to the same class. Note that the base and novel set are non-overlapping (i.e., $D_{base} \cap D_{novel} = \emptyset$).

**Overview.** As shown in Fig. 1, our MetaUAS framework is mainly composed of an encoder, a feature alignment module, and a decoder. The encoder extracts hierarchical features using a pre-trained model, while the decoder integrates query and prompt features to predict change heatmap. The feature alignment module is a bridge between the encoder and the decoder. It aligns the query and normal prompt features to address the geometric variation in spatial position. Concretely, for a query image $X^q$ and its corresponding prompt image $X^p$ ($X^p$ and $X^q \in \mathcal{R}^{H \times W \times 3}$), we parallelly extract

muti-scale offline features, $\{F_l^q\}_{l=1}^5$ and $\{F_l^p\}_{l=1}^5 \in \mathcal{R}^{h_l \times w_l \times c_l}$ from the encoder, where $l$ denote the $l$-th stage of the encoder. Then, the feature alignment module independently processes $F_l^q$ and $F_l^p$ for aligning query and prompt in feature space at each scale. Next, these aligned features are contacted and fed into the decoder for predicting change heatmaps. The MetaUAS is trained in a meta-learning manner. At inference, given a query image and its normal image prompt, the anomaly mask can be directly predicted. Next, we elaborately introduce them in this section.

**Encoder.** MetaUAS is compatible with any hierarchical architecture. Considering efficiency, we use the standard convolution-based EfficientNet-b4 [57] as our encoder following UniAD [70]. Given a query image $X^q$ and its prompt $X^p$, we extract multi-scale features $F_l^q$ and $F_l^p$ from the $l$-$th$ ($l = 1, 2, \cdots, 5$) stage of a pre-trained encoder $\mathcal{F}(\cdot; \theta_e)$, that is

$$F_l^q = \mathcal{F}(X^q; \theta_e), F_l^p = \mathcal{F}(X^p; \theta_e), \tag{1}$$

where $\theta_e$ is frozen to sufficiently utilize its generalization because it is pre-trained on a large-scale ImageNet.

**Feature Alignment Module.** We expect to learn a comparison between query and prompt features ($F_l^q$ and $F_l^p$) for improving change segmentation. A simple and naive manner is to directly contact them along the channel dimension, that is

$$F_l = \text{Concat}(F_l^q, F_l^p). \tag{2}$$

Then, the fused features $F_l$ are fed into a decoder to perform change segmentation. This simple fusion manner may only work when the query and its prompt are aligned in pixel space. However, this usually does not hold in practical applications because it is hard to refrain from geometric variations between query and prompt. Therefore, we have to align query and prompt features for better change segmentation. We propose two alignment strategies, hard and soft alignment, for enabling the interaction between query and prompt features.

**Hard Alignment** aims to search the most similar prompt feature at a spatial dimension for each query feature. Here, we take cosine similarity $\langle \cdot \rangle$ as a distance measure. Formally, for any query feature $F_l^p(i, j) \in \mathbb{R}^{c_l}$, the most similar prompt feature is

$$F_l^p(i, j) \leftarrow F_l^p\left( \underset{k,l}{\arg\min} \langle F_l^q(i, j), F_l^p(k, l) \rangle \right), \tag{3}$$

where $(i, j)$ and $(k, l)$ denote spatial locations. Considering computational efficiency, we use a $1 \times 1$ convolution layer $\text{Conv}(\cdot; \theta_a)$ with shared parameters $\theta_a$ to reduce the dimension of the channel before computing the cosine similarity, that is

$$F_l^{q\prime} \leftarrow \text{Conv}(F_l^q; \theta_a), F_l^{p\prime} \leftarrow \text{Conv}(F_l^p; \theta_a). \tag{4}$$

**Soft Alignment** is different from the hard alignment, which aligns each query feature with a weighted combination on the prompt feature. Similar to the hard alignment, we first apply Eq. 4 to reduce computation. The weighted probability is computed with the softmax function on a cross-similarity between the query and prompt features, that is

$$W_{ijkl} = \text{Softmax}\left( F_l^q(i, j)(F_l^p(k, l))^T \right), \tag{5}$$

where the $\text{Softmax}$ operation is applied to the last two dimensions, and thus $\sum_k \sum_l W_{ijkl} = 1$. Finally, the aligned prompt feature can be obtained by the weight and original prompt feature, that is

$$F_l^p(i, j) \leftarrow \sum_k \sum_l W_{ijkl} F_l^p(k, l). \tag{6}$$

The hard and soft alignment can adaptively align prompt features with query features, and thus handing geometric variation between query and prompt images to some extent. Appling Eqs. 3 or 6, we obtain an aligned prompt feature and then replace the original prompt feature $F_l^p$ in Eq. 2 with it.

**Decoder.** Considering efficiency, we apply the feature alignment module to three high-level features of prompt and query, and thus three fusion features $\{F_l\}_{l=3}^5$ are derived. Change segmentation needs to predict each pixel to determine whether it is changed. Specifically, we utilize UNet [46] $\mathcal{G}(\cdot; \theta_g)$ as our decoder because it is better suited for tasks requiring high precision and the preservation of fine-grained details. The UNet integrates all three fused features and two low-level original features and produces a final feature at the original image resolution. Finally, a segmentation head transforms the final feature to generate pixel-level change prediction $\hat{Y}$. The segmentation head is implemented by a simple $1 \times 1$ convolution layer, $\text{Conv}(\cdot; \theta_h)$, following a sigmoid activation.

## 3.3 Synthesizing Change Segmentation Images

Remote sensing [7] and street scenes [1] are two main scenarios in change segmentation. They mainly focus on semantic change, and various noises are included in unchanged background regions. Furthermore, the dataset scale is small and the diversity is insufficient. Therefore, it is not suitable for universal change segmentation. Recent works [64, 63, 22] have exploited generative diffusion models to create synthetic datasets and presented a promising performance on real datasets. However, it is hard to guarantee generative annotations are accurate. In contrast, some works have shown that simple synthesis, such as copy-paste, also brings strong performance for instance segmentation [19, 51] and anomaly segmentation [31, 72]. Similar to these works, we want to leverage a synthetic change segmentation dataset with accurate change masks. As early discussed, there are three main change types, "appearance", "disappearance", and "exchange", where "appearance" and "disappearance" are a pair of opposite concepts, and they can be transformed into each other by swapping paired images. Therefore, we only need to synthesize two change types to simulate all three ones.

**Object-Level Change.** In the famous MS-COCO [35], instances are annotated with polygons, and thus their foreground masks are available. Following CYWS [50], given a random instance and its binary mask from an image, we could make it disappear from the image by inpainting the mask region [56]. In this simple manner, we can simulate the "disappearance" change. Meanwhile, the change mask is freely available. The "appearance" change can be easily obtained by swapping original and unpainted images. It is challenging to synthesize the "exchange" change because two different instances usually mean different masks. But we can randomly paste an or multiple instances to a given image for rough simulation.

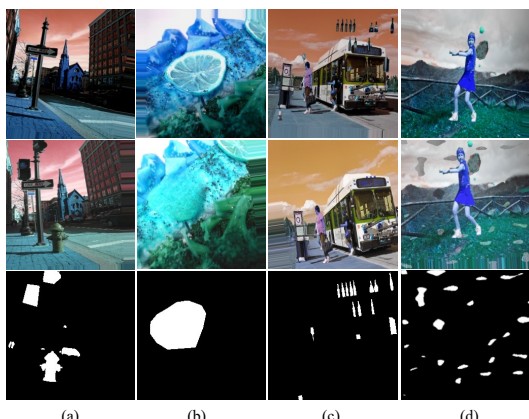

(a)  (b)  (c)  (d)

Figure 2: Selected synthesizing image pairs and their change masks. (a) and (b) simulate "appearance" and "disappearance" synthesizing with mask inpainting [56], and [50], (c) simulate "exchange" synthesizing with random pasting, and (d) simulate local region changes synthesizing with DRAEM [72].

**Local-Region Change.** Object-level changes are generated only by inpainting mask regions or randomly pasting objects as shown in Fig 2. However, anomaly changes are usually diverse, and they may be whole objects or local regions. The local "disappearance" and "appearance" may be failed by inpainting because local regions are easily restored by context. Following DRAEM [72], we first generate a binary change mask with Perlin noise and then synthesize a new image by filling the mask region with another image pixels. This synthesis can simulate local changes since the binary mask is generated randomly. Given a changed image pair, we apply various data augmentations, such as scale transformations, translation, rotation, and color jittering to enhance the diversity of changes during training phase.

## 3.4 Training and Inference

**Traning.** We train MetaUAS in a meta-learning manner, and each meta-task $\{X_i^p, X_i^q, Y_i\}$ is one prompt-query pair. The binary cross-entropy loss is adopted to optimize the learnable parameters ($\theta_a$, $\theta_g$, and $\theta_h$) of MetaUAS, that is

$$\mathcal{L} = -\sum_i \left( Y_i \cdot \log(\hat{Y}_i) + (1 - Y_i) \cdot \log(1 - \hat{Y}_i) \right). \tag{7}$$

**Inference.** For a class-specific query image $X^q$, we first randomly select a normal image prompt $X^p$ from the corresponding normal training set and then process prompt-query pairs online to perform anomaly segmentation. For a class-agnostic query image, we need to first construct a class-aware prompt pool $\{P_i\}_{i=1}^C$ via extracting offline features of all normal prompts $\{X_i^p\}_{i=1}^C$ in a total of $C$ classes, and then derive the best matching prompt by computing the cosine similarity between the query feature $F$ and the prompt pool. Here, the query and prompt features are obtained by using a global average pooling on the last stage feature from the encoder.

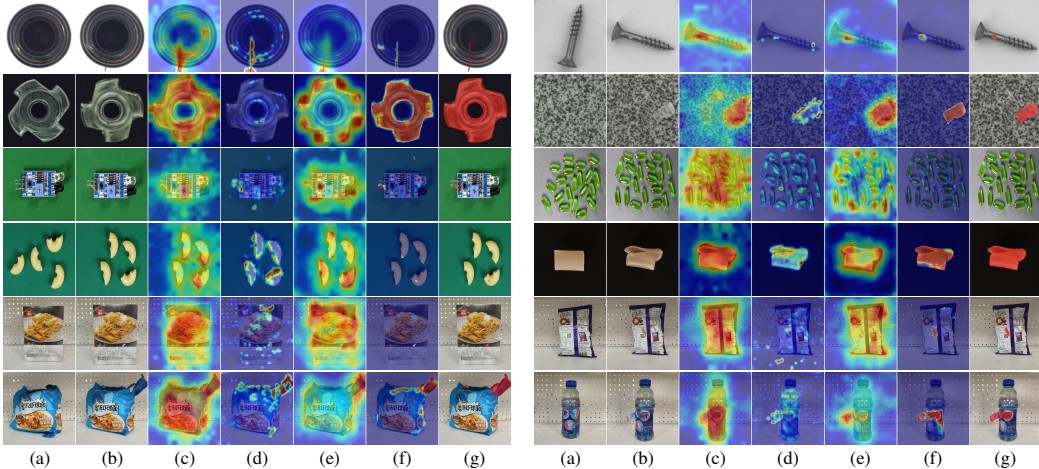

|  (a) | (b) | (c) | (d) | (e) | (f) | (g) | | (a) | (b) | (c) | (d) | (e) | (f) | (g) |

Figure 3: Qualitative comparisons with state-of-the-art methods on MVTec, VisA and Goods. In both two sub-figures (left and right), (b) and (g) represent query images and their anomaly masks, while (a) represent the corresponding normal image prompts. The predicted anomaly maps are shown using different methods, including (c) WinCLIP+ [26], (d) AnomalyCLIP [76], (e) UniAD [70] and (f) our MetaUAS. Best viewed in color and zoom-in.

Given a query image $X^q$ and the corresponding prompt $X^p$, we successively feed them into the encoder, the feature alignment module, the decoder, and the segmentation head, and finally obtain a predicted anomaly map $\hat{Y}$. Following previous works, we take the maximum of $\hat{Y}$ as the image-level anomaly score, without additional post-processing.

## 4 Experiment

### 4.1 Experimental Setup

Following previous works, we comprehensively evaluate MetaUAS on three industrial anomaly segmentation benchmarks, MVTec [4], VisA [78] and Goods [74]. We train a universal change segmentation model on a synthetic dataset. To demonstrate cross-domain generalization ability, we directly test it on three industry anomaly detection benchmarks without fine-tuning

**Competing Methods.** We compare **MetaUAS** and its two variants (**MetaUAS⋆** and **MetaUAS⋆+**) with diverse state-of-the-art anomaly segmentation methods including zero-shot CLIP [42], WinCLIP [26], AnomalyCLIP [76], and one-shot PatchCore [47], WinCLIP+ [26], and full-shot UniAD [70]. **MetaUAS** segments any anomalies with only one normal image prompt. Here, the one normal prompt is randomly sampled from normal training images for each class. The results are mean and standard deviation based on 5 independent repeated tests with different random seeds. **MetaUAS⋆** takes the best-matched normal image from the normal training set as the prompt of query image. Here, the matching degree is computed using the cosine similarity between the query image and all normal training images in feature space. **MetaUAS⋆+** builds on MetaUAS⋆. Following WinCLIP+ [26], we also add the visual prior knowledge from the image encoder of CLIP model to our MetaUAS⋆ for a fair comparison. The visual prior knowledge used in MetaUAS⋆+ is kept exactly the same as WinCLIP+.

**Evaluation Metrics.** Following previous works, we use ROC, PR, and $F1_{max}$ metrics for image-level anomaly classification. Similar to the anomaly classification evaluation, we use the same metrics and additionally report Per-Region Overlap (PRO) for pixel-level anomaly segmentation. We argue that the PR and $F1_{max}$ metrics are better for anomaly segmentation, where the imbalance issue is very extreme between normal and anomaly pixels [11, 78].

### 4.2 Comparison with Previous Works

Tables 1 and 2 present the comparison results of MetaUAS with the above-mentioned competing methods in generalization and efficiency, respectively.

Table 1: Quantitative comparisons on **MVTec**, **VisA** and **Goods**. Red indicates the best performance, while blue denotes the second-best result. Gray indicates the model is trained by full-shot normal images.

| Datasets | Methods | Venue | Shot | Auxiliary | Anomaly Classification | | | Anomaly Segmentation | | | |
|---|---|---|---|---|---|---|---|---|---|---|---|
| | | | | | I-ROC | I-PR | I-F1$_{max}$ | P-ROC | P-PR | P-F1$_{max}$ | P-PRO |
| MVTec | CLIP [42] | ICML 21 | 0 | ✗ | 74.4 | 89.3 | 88.7 | 62.0 | 6.5 | 11.2 | 21.4 |
| | PatchCore [47] | CVPR 22 | 1 | ✗ | 79.0±0.8 | 89.6±1.1 | 88.9±0.3 | 93.1±0.2 | 37.1±0.9 | 42.2±0.8 | 82.7±0.5 |
| | WinCLIP [26] | CVPR 23 | 0 | ✗ | 90.4 | 95.6 | 92.7 | 82.3 | 18.2 | 24.8 | 61.9 |
| | WinCLIP+ [26] | CVPR 23 | 1 | ✗ | 92.8±1.2 | 96.4±0.7 | 93.8±0.5 | 93.5±0.2 | 38.4±1.2 | 42.5±1.0 | 83.9±0.4 |
| | AnomalyCLIP [76] | ICLR 24 | 0 | ✓ | 91.5 | 96.3 | 92.7 | 91.1 | 34.5 | 39.1 | 81.4 |
| | UniAD [70] | NeurIPS 22 | full | ✗ | 96.7 | 98.9 | 96.7 | 96.8 | 44.7 | 50.4 | 90.0 |
| | **MetaUAS** | | 1 | ✗ | 90.7±0.7 | 95.7±0.6 | 92.5±0.3 | 94.6±0.2 | 59.3±1.4 | 57.5±1.1 | 82.6±0.6 |
| | **MetaUAS⋆** | | 1 | ✗ | 94.2 | 97.6 | 93.9 | 95.3 | 63.7 | 61.6 | 83.1 |
| | **MetaUAS⋆+** | | 1 | ✗ | 95.3 | 97.9 | 94.6 | 97.6 | 67.0 | 62.9 | 92.5 |
| VisA | CLIP [42] | ICML 21 | 0 | ✗ | 59.1 | 67.4 | 74.5 | 56.5 | 1.8 | 3.6 | 22.4 |
| | PatchCore [47] | CVPR 22 | 1 | ✗ | 64.2±1.0 | 66.0±0.7 | 75.5±0.5 | 95.5±0.3 | 16.5±1.7 | 26.0±1.5 | 84.6±0.5 |
| | WinCLIP [26] | CVPR 23 | 0 | ✗ | 75.5 | 78.7 | 78.2 | 73.2 | 5.4 | 9.0 | 51.0 |
| | WinCLIP+ [26] | CVPR 23 | 1 | ✗ | 80.5±2.6 | 82.1±2.7 | 81.3±1.0 | 94.4±0.1 | 15.9±0.2 | 23.2±0.4 | 79.3±0.3 |
| | AnomalyCLIP [76] | ICLR 24 | 0 | ✓ | 81.9 | 85.4 | 80.7 | 95.5 | 21.3 | 28.3 | 86.8 |
| | UniAD [70] | NeurIPS 22 | full | ✗ | 90.8 | 93.2 | 87.8 | 98.5 | 34.3 | 39.1 | 84.8 |
| | **MetaUAS** | | 1 | ✗ | 81.2±1.7 | 84.5±1.4 | 80.2±0.7 | 92.2±0.7 | 42.7±0.8 | 44.7±0.6 | 60.4±1.5 |
| | **MetaUAS⋆** | | 1 | ✗ | 83.4 | 85.7 | 81.3 | 92.0 | 43.9 | 45.6 | 57.3 |
| | **MetaUAS⋆+** | | 1 | ✗ | 85.1 | 87.2 | 82.3 | 98.0 | 48.1 | 48.6 | 85.5 |
| Goods | CLIP [42] | ICML 21 | 0 | ✗ | 51.8 | 57.3 | 71.3 | 55.3 | 4.3 | 2.0 | 16.4 |
| | PatchCore [47] | CVPR 22 | 1 | ✗ | 48.3±1.0 | 54.2±0.5 | 71.3±0.1 | 84.3±0.5 | 4.5±0.2 | 9.3±0.3 | 55.6±1.0 |
| | WinCLIP [26] | CVPR 23 | 0 | ✗ | 52.2 | 58.2 | 71.4 | 73.0 | 5.0 | 10.2 | 44.5 |
| | WinCLIP+ [26] | CVPR 23 | 1 | ✗ | 53.5±0.2 | 58.6±0.2 | 71.5±0.1 | 85.5±0.6 | 5.7±0.4 | 11.3±0.5 | 56.6±1.2 |
| | AnomalyCLIP [76] | ICLR 24 | 0 | ✓ | 57.2 | 63.3 | 71.4 | 83.5 | 16.9 | 24.0 | 63.3 |
| | UniAD [70] | NeurIPS 22 | full | ✗ | 67.5 | 72.1 | 74.6 | 90.4 | 15.0 | 20.6 | 66.1 |
| | **MetaUAS** | | 1 | ✗ | 54.5±1.0 | 58.5±0.4 | 71.5±0.1 | 88.5±0.6 | 8.6±0.7 | 14.0±0.7 | 59.0±1.3 |
| | **MetaUAS⋆** | | 1 | ✗ | 90.1 | 91.7 | 85.7 | 97.4 | 53.7 | 55.5 | 70.8 |
| | **MetaUAS⋆+** | | 1 | ✗ | 89.9 | 89.9 | 86.2 | 97.9 | 49.0 | 55.8 | 88.0 |

**Generalization.** First, MetaUAS with one normal image prompt achieves competitive performance among all zero-, few- and full-shot methods both on MVTec and VisA. This suggests that it is possible to boost anomaly classification and segmentation performance only with visual information alone. But on Goods dataset, MetaUAS seems to perform similarly to other methods. Different from MVTec and VisA, Goods consists of six groups and each group contains dozens or even hundreds of subcategories (484 in total). In fact, it is challenging to address multi-classes with a single model, and the state-of-the-art UniAD is not good even using all normal training images. In contrast, MetaUAS only uses one normal image prompt for each group, which means that the prompt image does not match most query images from multiple subcategories. Furthermore, MetaUAS⋆ significantly outperforms almost all competing models when the best-matched normal image is used to take as the normal prompt of each query image. In addition, WinCLIP+ boosts few-shot AS with dense similarity between few-shot prompts and query images. For a fair comparison, we also add the visual prior knowledge from the image encoder of CLIP model to our MetaUAS⋆ (denoting as MetaUAS⋆+). We can see that the performance can be further improved when introducing the visual prior of CLIP models to MetaUAS⋆.

**Efficiency.** We measure complexity and efficiency with the number of parameters and forward inference times. The evaluation is performed on one V100 GPU with batch size 32. The number of parameters of WinClIP+ and AnomalyCLIP is 10× and 20× of MetaUAS due to the large vision-language backbone. Compared to state-of-the-art, our MetaUAS⋆+ achieves the best performance using the single model with half of the parameters and faster inference time. What is more, our MetaUAS⋆ has 10× fewer parameters and 100× speed improvement compared to WinCLIP+, which still performs better.

**Qualitative Comparisons.** Figs. A1 and 4 show some selected visualizations from MVTec, VisA, and Goods testing images using the state-of-arts and our MetaUAS. Generally, MetaUAS segments anomalies more accurately and produces fewer false positives. MetaUAS is robust to different image prompts from the same category, especially for those categories with large geometric variations, such as screw. We believe that better performance can be derived if the objects or textures of the prompt image and query images can be roughly aligned.

Table 2: The complexity and efficiency comparisons. The performance of anomaly classification and segmentation is reported on **MVTec**.

| Methods | Backbone | #All Params(#Leanable) | Input Size | Times (ms) | I-ROC | P-ROC | P-PR |
|---|---|---|---|---|---|---|---|
| CLIP [42] | ViT-B-16+240 | 208.4 (0.0) | 240×240 | 13.7 | 74.4 | 62.0 | 6.5 |
| PatchCore [47] | E-b4 | 17.5 (0.0) | 256×256 | 36.4 | 79.0±0.8 | 93.1±0.2 | 37.1±0.9 |
| | | | 512×512 | 145.1 | 79.1±0.7 | 93.1±0.2 | 37.5±1.2 |
| WinCLIP [26] | ViT-B-16+240 | 208.4 (0.0) | 240×240 | 201.3 | 90.4 | 82.3 | 18.2 |
| WinCLIP+ [26] | | | | 339.5 | 92.8±1.2 | 93.5±0.2 | 38.4±1.2 |
| AnomalyCLIP [76] | ViT-L/14@336px | 433.5 (5.6) | 518×518 | 154.9 | 91.5 | 91.1 | 34.5 |
| UniAD [70] | Eb4 | 27.1 (7.7) | 224×224 | 5.0 | 96.7 | 96.8 | 44.7 |
| **MetaUAS** | Eb4 | 22.1 (4.6) | 256×256 | 3.1 | 90.7±0.7 | 94.6±0.2 | 59.3±1.4 |
| **MetaUAS⋆** | | | | | 94.2 | 95.3 | 63.7 |
| **MetaUAS⋆+** | Eb4+ViT-B-16+240 | 139.3 (4.6) | | 204.8 | 95.3 | 97.6 | 67.0 |
| **MetaUAS** | Eb4 | 22.1 (4.6) | 512×512 | 12.0 | 90.4±1.8 | 92.9±0.4 | 57.2±1.9 |
| **MetaUAS⋆** | | | | | 93.2 | 93.3 | 59.8 |
| **MetaUAS⋆+** | Eb4+ViT-B-16+240 | 139.3 (4.6) | | 213.0 | 94.8 | 97.1 | 65.8 |

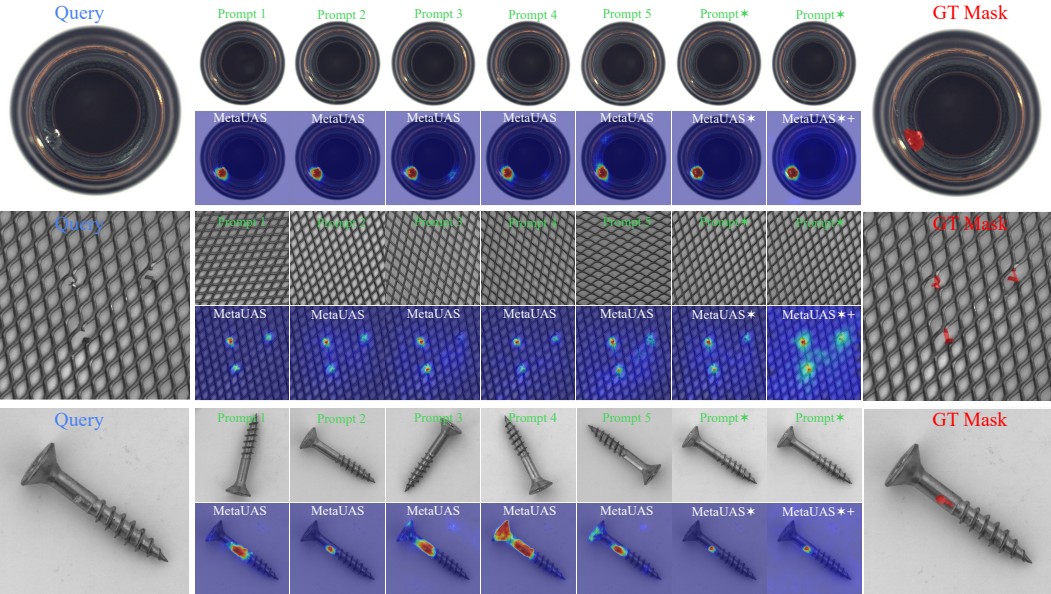

Figure 4: Anomaly segmentation for query images with different normal image prompts including 5 random prompts and the optimal prompt (denoting as prompt⋆). The anomaly segmentation maps are generated with MetaUAS, MetaUAS⋆ and MetaUAS⋆+.

## 4.3 Ablation Study

We perform component-wise analysis on MVTec with 256×256 inputs.

**The influence of feature alignment module.** As reported in Tab. 3a, we conduct experiments with combinations of different align strategies and feature fusions. The experimental results demonstrate that using soft alignment is better than the other two ones (no alignment and hard alignment) in the comprehensive results of AC and AS. Moreover, we compare the feature concatenation (Concat) with element-wised addition (Add) and absolute difference (AbsDiff) for aggregating prompt and query features. The Concat operation is the best one. The Add is not suitable to fuse two types of features as it fails to depict the image changes, and it may lead to confusion between these two features, as its results are the worst. The AbsDiff is widely used in the change segmentation field. However, this method may result in the loss of contextual information. In contrast, direct concatenation preserves all information and allows the network to adaptively learn the fusion, yielding the best results.

**The effects of change types.** The diversity of synthetic data is critical for good generalization. We use object-level changes and local region changes to ensure this diversity. Experiments show that object-level changes contribute more performance than local changes in Tab. 3c. This is natural because the object-level change is implemented by inpainting object regions and randomly pasting

Table 3: **Ablation studies on MVTec**. Default settings are marked in blue.

(a) Effect of feature alignment module.

| No. | Align | Fusion | I-ROC | I-PR | P-ROC | P-PR | P-PRO |
|---|---|---|---|---|---|---|---|
| 1 | No | Concat | 82.8 | 92.5 | 88.4 | 44.9 | 67.5 |
| 2 | Hard | Concat | 87.1 | 94.7 | 90.7 | 48.2 | 77.0 |
| 3 | Soft | Concat | **91.3** | **96.2** | **94.6** | **59.6** | **82.6** |
| 4 | Soft | Add | 71.8 | 86.9 | 73.2 | 24.0 | 45.2 |
| 5 | Soft | AbsDiff | 84.1 | 92.4 | 88.4 | 45.9 | 68.4 |

(b) Learn or freeze encoder?

| No. | Backbone | Learn? | I-ROC | I-PR | P-ROC | P-PR | P-PRO |
|---|---|---|---|---|---|---|---|
| 1 | E-b4 | Learn | 86.5 | 93.6 | 93.1 | 50.3 | 74.6 |
| 2 | E-b4 | Freeze | **91.3** | **96.2** | 94.6 | **59.6** | **82.6** |
| 3 | E-b6 | Freeze | 90.1 | 95.5 | 95.1 | 56.9 | 80.8 |
| 4 | EViT-b3 | Freeze | 89.5 | 95.7 | **95.3** | 58.5 | 80.9 |
| 5 | M-v2 | Freeze | 76.2 | 87.8 | 87.6 | 33.7 | 61.0 |

(c) Effects of change types and decoder module.

| No. | ChangeType | Decoder | I-ROC | I-PR | P-ROC | P-PR | P-PRO |
|---|---|---|---|---|---|---|---|
| 1 | Only Loc. | UNet | 83.1 | 92.8 | 87.7 | 44.3 | 76.1 |
| 2 | Only Obj. | UNet | 90.5 | 96.0 | 94.5 | 58.3 | 75.4 |
| 3 | Obj.+Loc. | UNet | **91.3** | **96.2** | **94.6** | **59.6** | **82.6** |
| 4 | Obj.+Loc. | FPN-Cat | 86.9 | 86.9 | 91.6 | 49.9 | 76.7 |
| 5 | Obj.+Loc. | FPN-Add | 88.4 | 94.7 | 94.1 | 51.4 | 73.1 |

(d) Effects of the number of training samples.

| No. | #Samples | I-ROC | I-PR | P-ROC | P-PR | P-PRO |
|---|---|---|---|---|---|---|
| 1 | 10% | 82.0 | 91.9 | 85.4 | 36.5 | 62.1 |
| 2 | 30% | 87.4 | 93.6 | 89.1 | 50.6 | 73.8 |
| 3 | 50% | 91.0 | 96.2 | 92.9 | 57.1 | 74.3 |
| 4 | 70% | 91.1 | **96.4** | 94.5 | 57.0 | 78.3 |
| 5 | 95% | **91.3** | 96.2 | **94.6** | **59.6** | **82.6** |

objects, which has a larger space than local region synthesis. Undoubtedly, combining them further enhances the diversity of synthetic changes, thus further improving performance.

**Learn or freeze encoder.** In Tab. 3b, if we train the encoder like other modules, the performance drops on both AC and AS. We speculate that it makes the network overfit change segmentation dataset, which will degenerate generalization. We also evaluate other backbones, such as EfficientNet-b6 [57], EfficientViT-b3 [6], and MobileNetV2 [53]. The EfficientNet-b4 performs better than others. The reason might be that shallow networks cannot extract discriminative features, while deep networks focus more on semantic features. AS requires more structural and texture features.

**The effects of decoder module.** In the decoder, we compare UNet [46] with FPN [34]. U-Net has been widely used and validated in many segmentation tasks due to its effectiveness and efficiency, while FPN is a type of network designed for object detection that requires recognizing objects at various scales. As reported in Tab. 3c, both different types of FPN are worse than UNet.

**The effects on training samples scale.** To investigate the influence of training samples scale on model performance, we conduct experiments with different training subsets where each one is generated by randomly sampling the original training set at various rates, such as $\{10\%, 30\%, 50\%, 70\%, 95\%\}$. The performance of each model on the MVTec testing set is reported in Tab. 3d. It can be seen that MetaUAS still works when the number of training images is small scale (e.g., 50%), and the performance can further improve when increasing the number of training samples.

## 5 Conclusion

This paper is the first study to focus on universal anomaly segmentation using pure visual information, enabling the segmenting of unseen anomalies without any training on target anomaly datasets or reliance on language guidance. First, we rethink anomaly segmentation tasks and find they can be unified into change segmentation. This paradigm shift allows us to break away from the persistent challenge of lacking large-scale anomaly segmentation datasets. Naturally, we are capable of leveraging large-scale synthetic image pairs with object-level and local region changes derived from available image datasets. Second, we propose a simple but effective universal anomaly segmentation framework, i.e., MetaUAS. We train MetaUAS in a one-prompt meta-learning manner on this synthesized dataset. To handle geometrical variations between prompt and query images, we propose a soft feature alignment module that bridges paired-image change segmentation and singe-image semantic segmentation. This makes it possible to use sophisticated semantic segmentation modules for change segmentation. MetaUAS achieves a superior generalization using only one normal image prompt on three industrial datasets. Meanwhile, MetaUAS enjoys faster inference speed and fewer parameters. We believe MetaUAS will serve as an alternative to widely used vision-language models for universal anomaly segmentation. **Limitation.** The performance of MetaUAS can be affected by using inappropriate normal image prompts. In this study, we leverage cosine similarity to identify the most suitable prompts when the category of the query image is unknown. In scenarios involving fine-grained objects, it may be essential to train a classification model to accurately predict the categories of the query images.

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

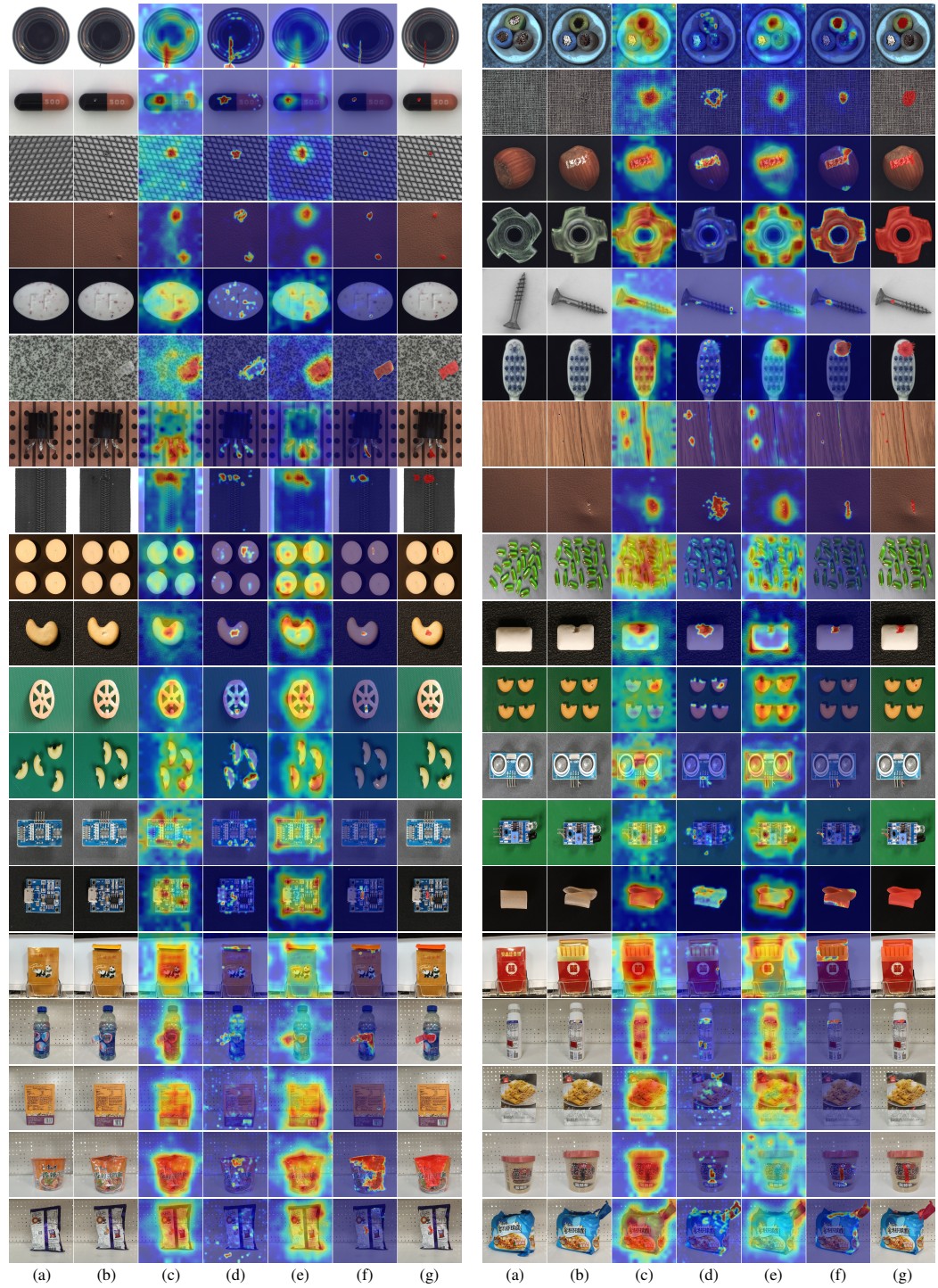

(a)    (b)    (c)    (d)    (e)    (f)    (g)        (a)    (b)    (c)    (d)    (e)    (f)    (g)

Figure A1: Qualitative comparisons with state-of-the-art methods on MVTec, VisA and Goods. In both two sub-figures (left and right), (b) and (g) represent query images and their anomaly masks, while (a) represent the corresponding normal image prompts. The predicted anomaly maps are shown using different methods, including (c) WinCLIP+ [26], (d) AnomalyCLIP [76], (e) UniAD [70] and (f) our MetaUAS. Best viewed in color and zoom-in.

Table A1: Quantitative results on **MVTec** with MetaUAS, MetaUAS⋆ and MetaUAS⋆+.

| Methods | Categories | Anomaly Classification | | | Anomaly Segmentation | | | |
|---|---|---|---|---|---|---|---|---|
| | | I-ROC | I-PR | I-F1$_{max}$ | P-ROC | P-PR | P-F1$_{max}$ | P-PRO |
| **MetaUAS** | bottle | 98.3±0.8 | 99.5±0.2 | 97.9±0.8 | 97.6±1.6 | 85.9±2.6 | 77.9±1.5 | 94.4±1.5 |
| | cable | 90.8±1.5 | 95.1±0.9 | 86.5±1.8 | 95.2±0.4 | 64.1±1.3 | 63.0±1.6 | 85.8±1.8 |
| | capsule | 67.1±5.2 | 89.8±3.0 | 91.4±0.5 | 94.2±0.7 | 23.6±6.5 | 33.7±4.3 | 54.3±7.0 |
| | carpet | 99.8±0.3 | 99.9±0.1 | 99.3±0.7 | 97.4±0.4 | 73.7±1.5 | 68.1±1.7 | 94.4±0.4 |
| | grid | 94.6±1.2 | 98.1±0.5 | 92.7±1.3 | 89.0±1.2 | 25.1±2.7 | 33.8±1.6 | 70.0±2.9 |
| | hazelnut | 97.9±2.2 | 98.9±1.2 | 95.0±3.2 | 98.1±0.7 | 66.2±9.8 | 60.3±8.4 | 87.9±3.5 |
| | leather | 99.9±0.2 | 100.0±0.0 | 99.7±0.3 | 99.7±0.0 | 71.2±0.9 | 65.4±0.8 | 95.8±0.8 |
| | metal nut | 94.4±3.9 | 98.6±1.1 | 95.0±1.2 | 95.0±0.7 | 76.9±4.0 | 70.9±2.6 | 87.1±1.2 |
| | pill | 92.3±1.4 | 98.5±0.2 | 94.2±1.1 | 96.4±0.6 | 70.1±2.9 | 63.8±2.3 | 88.6±2.1 |
| | screw | 63.5±5.0 | 84.4±3.4 | 85.5±0.3 | 92.1±3.1 | 8.1±2.9 | 14.4±3.9 | 72.4±5.7 |
| | tile | 95.6±0.6 | 98.5±0.1 | 94.4±1.1 | 95.3±0.5 | 84.6±1.0 | 79.5±0.7 | 92.0±0.8 |
| | toothbrush | 92.2±1.8 | 97.2±0.8 | 91.5±2.9 | 98.9±0.2 | 70.2±1.6 | 69.2±0.4 | 81.0±3.6 |
| | transistor | 79.7±6.6 | 79.3±6.8 | 71.9±4.2 | 82.4±3.2 | 37.2±5.1 | 37.7±5.0 | 67.1±3.6 |
| | wood | 98.5±0.3 | 99.5±0.1 | 96.7±0.8 | 94.1±0.4 | 70.0±1.7 | 65.9±2.1 | 89.0±1.1 |
| | zipper | 95.9±2.5 | 98.5±1.3 | 96.1±1.2 | 94.5±1.3 | 62.1±2.2 | 59.5±1.7 | 78.7±2.2 |
| | mean | 90.7±0.7 | 95.7±0.6 | 92.5±0.3 | 94.6±0.2 | 59.3±1.4 | 57.5±1.1 | 82.6±0.6 |
| **MetaUAS⋆** | bottle | 99.6 | 99.9 | 98.4 | 97.5 | 85.6 | 77.5 | 95.4 |
| | cable | 95.3 | 97.6 | 91.9 | 96.3 | 67.5 | 65.9 | 90.2 |
| | capsule | 80.1 | 94.9 | 93.5 | 95.8 | 40.5 | 48.3 | 57.6 |
| | carpet | 99.6 | 99.9 | 98.9 | 97.0 | 73.9 | 68.7 | 93.2 |
| | grid | 96.2 | 98.7 | 94.8 | 90.8 | 28.7 | 37.1 | 75.6 |
| | hazelnut | 99.3 | 99.6 | 97.9 | 98.8 | 74.7 | 68.0 | 89.1 |
| | leather | 100 | 100 | 100 | 99.7 | 70.9 | 65.5 | 96.4 |
| | metal nut | 96.2 | 99.1 | 95.2 | 96.3 | 81.4 | 73.3 | 91.0 |
| | pill | 95.3 | 99.2 | 94.7 | 94.8 | 64.8 | 59.9 | 86.3 |
| | screw | 84.2 | 94.5 | 87.6 | 95.0 | 29.4 | 33.4 | 61.7 |
| | tile | 95.1 | 98.3 | 93.4 | 94.6 | 83.3 | 78.8 | 91.2 |
| | toothbrush | 93.6 | 97.6 | 92.3 | 98.9 | 70.3 | 70.5 | 78.6 |
| | transistor | 91.0 | 88.3 | 79.2 | 86.0 | 47.9 | 48.0 | 72.8 |
| | wood | 98.8 | 99.6 | 96.8 | 94.3 | 73.0 | 68.4 | 88.2 |
| | zipper | 89.3 | 96.3 | 93.7 | 94.2 | 63.7 | 61.5 | 79.0 |
| | mean | 94.2 | 97.6 | 93.9 | 95.3 | 63.7 | 61.6 | 83.1 |
| **MetaUAS⋆+** | bottle | 99.6 | 99.9 | 98.4 | 98.8 | 87.5 | 78.1 | 96.8 |
| | cable | 95.5 | 97.7 | 91.9 | 97.1 | 67.4 | 66.4 | 91.6 |
| | capsule | 83.4 | 95.7 | 92.7 | 97.8 | 43.3 | 49.4 | 90.0 |
| | carpet | 99.8 | 100 | 98.9 | 99.5 | 80.6 | 71.0 | 98.0 |
| | grid | 99.6 | 99.9 | 98.2 | 98.2 | 36.5 | 39.4 | 94.7 |
| | hazelnut | 100 | 100 | 100 | 99.1 | 79.1 | 74.1 | 92.7 |
| | leather | 100 | 100 | 100 | 99.7 | 71.6 | 65.5 | 98.9 |
| | metal nut | 97.8 | 99.5 | 96.3 | 96.5 | 82.1 | 73.7 | 92.1 |
| | pill | 95.8 | 99.3 | 95.0 | 96.8 | 68.5 | 60.9 | 94.1 |
| | screw | 88.2 | 95.6 | 91.3 | 98.4 | 34.4 | 33.9 | 90.5 |
| | tile | 96.1 | 98.6 | 94.0 | 98.1 | 88.4 | 79.3 | 95.4 |
| | toothbrush | 94.4 | 97.9 | 92.3 | 99.4 | 72.6 | 70.9 | 91.7 |
| | transistor | 91.1 | 88.5 | 80.5 | 91.6 | 51.0 | 50.6 | 78.6 |
| | wood | 99.0 | 99.7 | 96.8 | 96.7 | 77.4 | 69.9 | 95.0 |
| | zipper | 89.4 | 96.4 | 93.4 | 96.0 | 64.7 | 61.0 | 87.9 |
| | mean | 95.3 | 97.9 | 94.6 | 97.6 | 67.0 | 62.9 | 92.5 |

# A    Implementation Details.

Following UniAD [70], we extract multi-scale features from all 5 stages of EfficientNet-b4 [57] encoder. In the feature alignment module, the three highest-level features are used to perform query-prompt alignment, and the channel number is reduced to half of one of the original channels before calculating the similarity between query and prompt. Therefore, we derive three aligned features of query and prompt using the feature alignment module. Finally, these three aligned features and two original low-level query features from the first and second stages are fed into the decoder and segmentation head for change segmentation. The model is trained with 30 epochs on 8 Tesla V100 GPUs with batch size 128. We freeze the encoder and optimize the feature alignment module, the

Table A2: Quantitative results on **VisA** with MetaUAS, MetaUAS⋆ and MetaUAS⋆+.

| Methods | Categories | Anomaly Classification | | | Anomaly Segmentation | | | |
|---|---|---|---|---|---|---|---|---|
| | | I-ROC | I-PR | I-F1$_{max}$ | P-ROC | P-PR | P-F1$_{max}$ | P-PRO |
| **MetaUAS** | candle | 84.7±1.1 | 85.2±1.4 | 79.7±1.1 | 99.3±0.1 | 60.0±2.3 | 57.3±1.7 | 63.0±3.2 |
| | capsules | 77.7±3.9 | 86.4±2.3 | 79.7±1.7 | 96.5±0.7 | 40.5±4.1 | 44.8±3.4 | 76.9±2.5 |
| | cashew | 78.9±5.1 | 90.1±2.4 | 82.2±1.7 | 91.1±1.9 | 49.4±3.7 | 50.4±2.4 | 51.8±4.2 |
| | chewinggum | 95.8±0.2 | 98.2±0.1 | 93.5±1.1 | 98.5±0.4 | 85.2±1.5 | 79.6±1.1 | 69.6±0.9 |
| | fryum | 83.5±2.4 | 91.9±1.4 | 84.0±1.4 | 65.2±5.4 | 14.9±5.5 | 23.0±6.6 | 23.9±2.5 |
| | macaroni1 | 73.0±6.0 | 77.0±5.4 | 71.2±2.0 | 82.4±2.1 | 13.1±6.3 | 21.2±8.0 | 31.5±4.0 |
| | macaroni2 | 60.8±4.2 | 59.8±3.5 | 68.0±0.7 | 89.5±5.7 | 2.3±1.1 | 7.5±2.9 | 56.4±13.7 |
| | pcb1 | 75.4±13.6 | 76.0±9.8 | 75.1±9.6 | 98.2±0.6 | 66.1±5.8 | 62.9±4.1 | 71.4±6.3 |
| | pcb2 | 76.0±2.9 | 76.6±3.0 | 72.9±3.5 | 94.5±0.2 | 30.8±2.7 | 39.0±2.7 | 66.4±4.0 |
| | pcb3 | 77.1±3.4 | 79.8±2.6 | 72.8±2.7 | 97.0±0.4 | 42.7±3.4 | 42.9±1.9 | 58.5±1.5 |
| | pcb4 | 95.2±2.4 | 95.0±2.1 | 89.3±4.1 | 97.1±0.8 | 41.3±3.2 | 45.6±2.4 | 69.3±3.4 |
| | pipe fryum | 95.8±1.4 | 97.5±1.2 | 93.9±1.3 | 96.9±1.0 | 66.0±3.4 | 62.7±2.7 | 86.2±4.3 |
| | mean | 81.2±1.7 | 84.5±1.4 | 80.2±0.7 | 92.2±0.7 | 42.7±0.8 | 44.7±0.6 | 60.4±1.5 |
| **MetaUAS⋆** | candle | 84.4 | 85.4 | 78.8 | 98.9 | 59.8 | 57.5 | 55.9 |
| | capsules | 83.4 | 90.0 | 82.3 | 97.1 | 48.3 | 50.6 | 74.7 |
| | cashew | 84.3 | 92.1 | 85.6 | 88.8 | 43.5 | 45.6 | 48.8 |
| | chewinggum | 95.0 | 98.0 | 93.3 | 98.6 | 85.9 | 80.1 | 70.4 |
| | fryum | 84.1 | 92.8 | 83.4 | 67.1 | 13.7 | 20.6 | 22.4 |
| | macaroni1 | 71.6 | 74.3 | 71.1 | 81.0 | 4.7 | 10.4 | 24.6 |
| | macaroni2 | 60.3 | 57.9 | 67.6 | 91.0 | 2.8 | 9.6 | 65.1 |
| | pcb1 | 86.9 | 84.8 | 80.8 | 98.6 | 78.8 | 74.5 | 63.8 |
| | pcb2 | 79.9 | 78.7 | 75.0 | 95.9 | 34.9 | 41.0 | 64.5 |
| | pcb3 | 79.7 | 81.6 | 73.9 | 96.4 | 46.4 | 46.4 | 52.5 |
| | pcb4 | 96.1 | 95.3 | 91.1 | 95.4 | 43.7 | 46.9 | 62.8 |
| | pipe fryum | 95.6 | 97.8 | 92.5 | 95.1 | 64.8 | 63.5 | 82.6 |
| | mean | 83.4 | 85.7 | 81.3 | 92.0 | 43.9 | 45.6 | 57.3 |
| **MetaUAS⋆+** | candle | 85.8 | 86.3 | 79.8 | 98.3 | 58.5 | 57.5 | 92.9 |
| | capsules | 84.5 | 91.0 | 82.3 | 98.3 | 51.5 | 51.8 | 80.4 |
| | cashew | 87.7 | 93.5 | 88.9 | 98.5 | 55.9 | 50.6 | 88.1 |
| | chewinggum | 95.8 | 98.3 | 93.3 | 99.5 | 86.0 | 80.2 | 85.1 |
| | fryum | 89.6 | 94.9 | 88.2 | 96.6 | 38.6 | 44.5 | 81.9 |
| | macaroni1 | 73.1 | 76.3 | 70.8 | 96.9 | 7.8 | 12.5 | 81.1 |
| | macaroni2 | 62.6 | 64.4 | 67.6 | 97.7 | 4.6 | 10.5 | 89.6 |
| | pcb1 | 87.9 | 86.0 | 81.7 | 99.3 | 81.8 | 75.7 | 82.4 |
| | pcb2 | 80.4 | 79.1 | 75.4 | 97.4 | 35.1 | 41.8 | 77.4 |
| | pcb3 | 80.7 | 82.3 | 75.1 | 96.8 | 46.7 | 47.2 | 85.7 |
| | pcb4 | 96.6 | 95.8 | 91.6 | 97.2 | 43.6 | 47.1 | 84.3 |
| | pipe fryum | 96.5 | 98.3 | 93.0 | 99.0 | 66.9 | 63.5 | 96.6 |
| | mean | 85.1 | 87.2 | 82.3 | 98.0 | 48.1 | 48.6 | 85.5 |

decoder, and the segmentation head with AdamW [25] using weight decay 0.0005 and learning rate 0.0001. We conduct experiments based on the open-source framework PyTorch.

We follow CYWS [50] and use the same procedure for synthesizing the change segmentation dataset. Specifically, given a labeled image from an existing instance segmentation dataset, i.e., MS-COCO, we randomly selected one or several instances and then could make it disappear from the image by inpainting the mask region [56]. It is worth noting that the binary change mask between the inpainted and original images can be freely available because these selected instances have been manually annotated at the pixel level. We keep the dataset setup as similar to CYWS [50] as possible. Specifically, the change segmentation dataset is synthesized using the randomly selected 60,000 images from the MS-COCO training set. For each image, a synthesized image is generated by inpainting a union mask of a random set of labeled instances. Then, all these 60,000 samples are divided into training and validation sets with a ratio of 0.95:0.05. During training, we randomly employ object-level change and local-region change with a probability of 0.5.

## B  Competing Methods.

To demonstrate the superiority of MetaUAS, we compare MetaUAS and its variants (MetaUAS⋆ and MetaUAS⋆+) with diverse state-of-the-art methods. Implementation and reproduction details are summarized as follows:

Table A3: Quantitative results on **Goods** with MetaUAS, MetaUAS⋆ and MetaUAS⋆+.

| Methods | Categories | Anomaly Classification | | | Anomaly Segmentation | | | |
| | | I-ROC | I-PR | I-F1$_{max}$ | P-ROC | P-PR | P-F1$_{max}$ | P-PRO |
| --- | --- | --- | --- | --- | --- | --- | --- | --- |
| **MetaUAS** | cigarette box | 58.9±3.8 | 63.2±3.6 | 74.2±0.5 | 88.4±1.3 | 21.1±3.3 | 28.9±2.9 | 62.9±3.3 |
| | drink bottle | 55.1±1.2 | 59.3±1.1 | 70.6±0.2 | 92.4±0.7 | 7.3±1.6 | 12.8±2.0 | 64.6±0.7 |
| | drink can | 52.1±3.4 | 48.4±2.0 | 66.7±0.0 | 86.6±1.4 | 7.6±1.1 | 14.0±0.8 | 58.4±0.5 |
| | food bottle | 55.0±1.4 | 64.4±0.5 | 75.0±0.1 | 89.9±0.3 | 8.4±0.6 | 14.2±0.6 | 59.4±1.0 |
| | food box | 52.9±2.6 | 65.6±2.5 | 77.7±0.3 | 86.4±1.7 | 4.4±0.7 | 8.3±1.1 | 57.2±2.3 |
| | food package | 52.7±1.7 | 50.4±1.8 | 64.8±0.1 | 87.5±1.5 | 2.8±0.6 | 5.7±1.4 | 51.6±3.2 |
| | mean | 54.5±1.0 | 58.5±0.4 | 71.5±0.1 | 88.5±0.6 | 8.6±0.7 | 14.0±0.7 | 59.0±1.3 |
| **MetaUAS⋆** | cigarette box | 98.9 | 99.2 | 96.0 | 98.7 | 78.0 | 73.8 | 88.0 |
| | drink bottle | 85.2 | 86.7 | 80.9 | 98.9 | 62.2 | 61.3 | 68.1 |
| | drink can | 96.7 | 97.1 | 91.5 | 93.8 | 44.9 | 53.7 | 57.9 |
| | food bottle | 90.1 | 93.1 | 86.2 | 97.1 | 50.5 | 52.1 | 70.1 |
| | food box | 86.9 | 92.4 | 84.5 | 98.3 | 54.6 | 54.8 | 67.5 |
| | food package | 82.7 | 81.8 | 74.8 | 97.5 | 32.1 | 37.0 | 73.6 |
| | mean | 90.1 | 91.7 | 85.7 | 97.4 | 53.7 | 55.5 | 70.8 |
| **MetaUAS⋆+** | cigarette box | 97.5 | 96.3 | 96.4 | 98.6 | 74.9 | 74.0 | 95.3 |
| | drink bottle | 85.4 | 86.8 | 81.3 | 98.8 | 58.7 | 61.4 | 87.6 |
| | drink can | 97.2 | 97.5 | 91.8 | 96.7 | 42.8 | 54.9 | 86.5 |
| | food bottle | 90.4 | 92.9 | 86.7 | 97.5 | 44.1 | 52.2 | 88.6 |
| | food box | 85.2 | 87.4 | 84.1 | 97.8 | 46.5 | 54.6 | 85.5 |
| | food package | 83.6 | 78.1 | 76.9 | 97.9 | 27.0 | 37.4 | 84.4 |
| | mean | 89.9 | 89.9 | 86.2 | 97.9 | 49.0 | 55.8 | 88.0 |

**CLIP** [42] is a powerful vision-language model, and it has a strong zero-shot generalization ability. Following previous works, we use two classes of text prompt templates, "A photo of a normal [cls]" and "A photo of an anomalous [cls]", where "cls" denotes the target class name. The anomaly score is computed by cosine similarity between textual features and the class token of a query image. For anomaly segmentation, we extend the above computation from class tokens to local patch tokens.

**WinCLIP** [26] is a zero-shot anomaly segmentation method based on CLIP. A large set of hand-crafted textual prompts is designed for anomaly classification. A window scaling strategy is used to obtain better anomaly segmentation. We keep all parameters the same as in their paper. Note that no official implementation of WinCLIP is available, our results are based on an unofficial implementation [1].

**WinCLIP+** [26] combines the complementary prediction from both language-guided and visual-based for better anomaly classification and segmentation. The language-guided prediction is the same as WinCLIP. For visual-based prediction, it first simply stores multi-scale features for given few-shot normal images and retrieves the memory features based on the cosine similarity. The final anomaly score is derived by averaging these two scores.

**AnomalyCLIP** [76] learns object-agnostic text prompts that capture generic normality and abnormality in an image regardless of its foreground objects. But AnomalyCLIP requires fine-tuning on an auxiliary domain dataset including normal and anomaly images. AnomalyCLIP is a zero-shot anomaly classification and segmentation method, and it is capable of recognizing any anomalies. We use the official model to report performance for anomaly classification and segmentation.

**UniAD** [70] is a unified unsupervised anomaly segmentation method for addressing multi-classes anomalies with a single model. Different from most zero-/few-shot anomaly segmentation models, UniAD learns feature reconstruction with a transformer-based encoder-decoder architecture on all normal training images. We use the official code to train the specific model for each dataset.

**PatchCore** [47] is a popular unsupervised anomaly classification method that enjoys training-free. For a fair comparison, we modify the official implementation in two folds. First, we replace the original WideResNet-50 backbone with EfficientNet-b4. Second, the memory-bank construction is limited to only one normal image for each class.

---

[1] `https://github.com/zqhang/Accurate-WinCLIP-pytorch`

