# OpenReview forum: "MetaUAS: Universal Anomaly Segmentation with One-Prompt Meta-Learning"
_NeurIPS.cc/2024/Conference — NeurIPS 2024 poster_

### Official Review · Reviewer_bbUz · 2024-06-25

**Soundness:** 3
**Presentation:** 3
**Contribution:** 3
**Rating:** 6
**Confidence:** 3

**Summary:**

This work presents a novel approach for universal anomaly segmentation (AS) by framing AS as Change segmentation. This change of perspective motivates them to create a synthetic training set based on change detection (e.g. by stitching additional objects from a different dataset). With this synthetic dataset they train a change segmentation model which they later transfer directly for anomaly segmentation without fine-tuning.

**Strengths:**

The idea of framing Anomaly Segmentation as Change Segmentation is interesting and worth exploring.
The results are encouraging, especially in the Gooods dataset.

**Weaknesses:**

The paper is confusing at times, with some important concepts not clearly defined:
    - For instance, the different method variants i.e. MetaUAS/MetaUAS*/MetaUAS*+ should be explained more thoroughly in the methodology section.
    - Also there should be consistency in the naming (MetaUAS vs MetaAS).
    - The highlighted numbers (red and blue) in the tables are excluding UniAD in Table 1 (which is marked in gray) and I could not find out why.

Missing comparison with change segmentation methods. The paper proposes that AS can be tackled as change segmentation between one normal (reference) image and the query image with the anomaly, which sounds reasonable. However, I would expect then that existing Change Segmentation methods could be applied to AS (just like authors do with their method). Thus, it would be sensible to include a comparison with one or two baselines of recent change segmentation methods applied to AS the same way that MetaUAS is applied.

**Questions:**

Can the authors explain what are the differences between MetaUAS/*/*+ ?

Why is UniAD excluded from the comparison in Table1?

Could all change segmentation methods be applied to anomaly segmentation "off-the-shelf"? If yes, why not include some existing methods as baseline? If no, please explain what are the particular differences that make MetaUAS applicable to AS while other change segmentation methods can not.

**Limitations:**

Limitations and social impact are well addressed.

---

> ### Author Rebuttal · Authors · 2024-08-05
>
> + **bbUz-Q1**: What are the differences between MetaUAS, MetaUAS*, and MetaUAS*+?
>
> We apologize for the confusion about MetaUAS, MetaUAS*, and MetaUAS*+. In fact, we have given some details of our MetaUAS and its two variants (MetaUAS* and MetaUAS*+) in the Sec. B of Supplementary Materials. Here, we provide some explanations, which might address your concerns.
>
> The MetaUAS is capable of segmenting any anomalies only given just one normal image prompt. Note that the normal image prompt is randomly selected for each testing class in MetaUAS. Different from MetaUAS, the MetaUAS* searches the best-matched normal image as a prompt from the normal training set for each query image. The MetaUAS*+ builds on MetaUAS*, but introduces the visual prior from a CLIP model. The visual prior knowledge is obtained by computing the cosine similarity between the query feature and the corresponding prompt feature extracted from the vision encoder of the CLIP model, which is the same as WinCLIP+ and PromptAD. In addition, the MetaAS is a typo and the correct one should be MetaUAS. We will add these clarifications to the methodology section.
>
> + **bbUz-Q2**: Why UniAD is excluded from the comparison in Table 1?
>
> UniAD is a powerful unsupervised anomaly detection method, which trains a unified reconstruction model to detect multi-class anomalies using full-shot normal images from the target domain. In contrast to our training-free one-shot method, UniAD requires re-training the model using full-shot normal images. As reported in Table 1, we observe that our one-shot training-free method still exhibits strong performance compared to the full-shot UniAD. We will add a description for UniAD in the caption of Tabel 1.
>
>
> + **bbUz-Q3**: Could all change segmentation methods be applied to anomaly segmentation ``off-the-shelf''?
>
> It is challenging to apply existing change segmentation models to anomaly segmentation. Current change segmentation models are primarily focused on two scenarios: remote sensing and street scenes. The main characteristics of these scenarios and models can be summarized in four aspects.
>
> First, the scale of training images in these scenarios is usually small, typically hundreds or thousands of images.
> Second, the diversity of images is limited because they mainly consist of remote sensing images or street scene images.
> Third, these models primarily focus on semantic changes (such as buildings) while the background may undergo other changes due to factors like seasons and weather.
> Fourth, the prompt and query images are generally coarsely aligned in the change segmentation, which is different from anomaly detection scenarios where there are often large geometric variations.
>
> These characteristics limit the development of a general change detection model. Additionally, we evaluated the [off-the-shelf ChangerEx with s50 backbone](https://github.com/likyoo/open-cd/tree/main/configs/changer) on MVTec, and the corresponding results are reported in the following table. We can see that its generalization performance is poor compared to our method.
>
>
> |Methods  |Backbone  |Training Data   |I-ROC  |I-PR  |P-ROC  |P-PR  |P-PRO|
> | :---  |    :----:   |  :----:  |  :----: |  :----: |  :----: |  :----: |  :----:  |
> |ChangerEx |S50  |LEVIR-CD          |59.8  |74.1   |65.5  |3.0    |0.8   |
> |ChangerEx |S50  |S2Looking	        |63.4  |79.6   |68.3  |10.5  |13.1 |
> |Ours	   |E-b4 |Synthesize	        |91.3  |96.2   |94.6  |59.6  |82.6 |

---

> > ### Comment · Reviewer_bbUz · 2024-08-09
> > **Thank you for the authors response**
> >
> > The author response clarified my questions and after reading the other reviewers comments I will increase my score to weak accept.

---

### Official Review · Reviewer_sm3a · 2024-07-06

**Soundness:** 3
**Presentation:** 2
**Contribution:** 2
**Rating:** 6
**Confidence:** 3

**Summary:**

The authors introduce a method for anomaly segmentation that relies solely on visual information and does not require any anomaly training data or language guidance. Their method is based on change segmentation, which allows for the synthesis of large-scale image pairs for training the anomaly segmentation model. The proposed MetaUAS framework demonstrates its generalizability across three industrial anomaly segmentation datasets: MVTec, VisA, and Goods, using one normal image per class as a prompt. Within their framework, the authors extensively experimented with different modules, including an image encoder, feature alignment, and a segmentation decoder. Notably, their feature alignment approach, which is based on a weighted combination of image prompt features, proves to be highly effective for anomaly segmentation by bringing together features of semantic segmentation and change segmentation.

**Strengths:**

- A novel approach is presented by formulating anomaly segmentation as change segmentation.
- Extensive experiments on multiple datasets and in different model configurations demonstrate the effectiveness of the presented method, MetaUAS. In particular, the anomaly segmentation performance benefits from the specifically designed soft alignment module.
- It is shown how efficiently a change segmentation dataset can be created with synthetic data, which is then used for training anomaly segmentation.
- MetaUAS, trained on synthetic data, generalizes well to real-world problems. Notably, the method does not require any anomalous data for training but still outperforms existing methods that use such auxiliary data.
- Unlike other existing methods, MetaUAS does not require guidance from language and solely relies on visual information.

**Weaknesses:**

- At least one normal image per class is always needed as a prompt to perform anomaly segmentation. All prompts must be processed by the encoder to extract their features, which could become expensive as the number of classes increases.
- The prompt features can be stored offline as a prompt pool to reduce forward passes at test time when using MetaUAS. However, it is unclear whether a single example can represent all normal examples for a class effectively (Figure 4 highlights the issue of different prompts). Searching for the best matching normal image for each query image is obviously resource-intensive.
- The anomalies shown in the industrial datasets relate to small changes in the appearance of the objects. The objects are also always displayed in the same scene. It is noted that the orientation of the object affects the anomaly segmentation performance. To further investigate the limitations, it would be interesting to see what would happen if the scenes changed more dramatically, which is realistic in open-world settings. For example, would MetaUAS be robust if the same normal object were shown with a different background?
- Minor: The paper could benefit from better writing and structure. There are a few typos throughout the text, and the figures and tables could be placed differently. It is not always easy to follow.

**Questions:**

- Some more details on the training process would be appreciated. For instance, there is no information on how much training data was required to achieve the presented performance or how robust the model training was. The training data seems to be created in an automated fashion; to what extent does the amount of training data affect the anomaly segmentation performance?
- One could use more than one image as a prompt to potentially obtain more robust anomaly scores. How efficient is the inference? In particular, compared to other existing approaches, could one afford a larger prompt pool or multiple forward passes of MetaUAS?
- Relying on the changes in the visual appearance of objects seems to be a valid approach for anomaly segmentation in industrial settings. The same objects often appear in the same scene, and the anomalies are clear, small visual changes. Therefore, guidance with language seems unnecessary, as the semantic information is not crucial for this kind of anomaly segmentation. However, for other datasets on anomaly segmentation, semantic information seems crucial, such as the [SMIYC](https://segmentmeifyoucan.com/) benchmark, where any obstacle on the road is considered an anomaly. Could MetaUAS also be applied to such more challenging tasks where the normal "object" itself (e.g., the road) could have varying visual appearances as well?

**Limitations:**

The authors have discussed limitations. Potential negative social impact have not been mentioned.

---

> ### Author Rebuttal · Authors · 2024-08-05
>
> + **sm3a-Q1**: The effects of the number of training data for the anomaly segmentation performance.
>
> Thanks for your interest in the effects of training scale. We are also very concerned about this issue, which has been analyzed in our ablation studies (Sec. 4.3). The corresponding experimental results are reported in Table 3(d), with the related analysis provided in Lines 361-366. The number of synthesized images and the split of training and validation are described in Sec. A of Supplementary Materials. Here, we provide further explanations.
>
> To investigate the influence of training scale on model performance, we conduct experiments with different training subsets where each one is generated by randomly sampling the original training set at various rates, such as {10%, 30%, 50%, 70%, 95%}. In Table 3(d), it can be seen that MetaUAS still works when the number of training images is small scale (e.g., 50%), and the performance can further improve when increasing the number of training samples. Unless otherwise specified, the default number of training images is 95% of the synthesized dataset.
>
> + **sm3a-Q2**: Compared to other existing approaches, could one afford a larger prompt pool or multiple forward passes of MetaUAS?
>
> We agree with you that using more normal image prompts can potentially enhance the robustness of the model. Additionally, we have observed that the performance gains tend to saturate when the number of normal image prompts increases to a certain level (e.g., 5) (see the response to gb5F-Q3). Assuming that each query image is provided with 5 normal image prompts, this means that the inference cost could increase by up to 5 times. However, considering that we use a lightweight model, even when 32 prompt-query pairs are processed in parallel during forward inference, the required time of each pair is only 3.2ms, as reported in Table 2. Therefore, our method can efficiently handle larger prompt pools compared to existing CLIP-based models.
>
> + **sm3a-Q3**: Could MetaUAS also be applied to such tasks where the normal ``object'' itself (e.g., the road, SMIYC) could have varying visual appearances as well?
>
> Our method is primarily towards open-world industrial anomaly detection, which has become an attractive yet challenging topic. It may not be suitable to directly apply our method in these scenarios where the normal ``object'' itself or its context exhibits large variations in appearance. This is because the core of our method relies on detecting changes to identify anomalies.
>
> Furthermore, we have carefully reviewed the suggested RoadAnomaly21, which contains only 100 testing images and does not provide normal images. In this dataset, the context changes of the road are more significant than that of the road itself. We believe that one feasible manner could be to first preprocess these images to remove backgrounds (i.e., the context of the road) and then apply our method to detect anomalies on the road surface.
>
> + **sm3a-Q4**: All prompts must be processed by the encoder to extract their features, which could become expensive as the number of classes increases.
>
> We apologize for any confusion this issue may have caused. Here, we provide some clarification that might address your concerns. For a class-specific query image, we can process prompt-query pairs online to perform anomaly segmentation without extracting offline features for normal image prompts. For a class-agnostic query image, we need to first extract offline features from a prompt pool, and then match the corresponding image prompt for the given query image to perform anomaly segmentation. In fact, in real-world industrial production lines, products are generally produced on a large scale according to specific models, so we could register normal image prompts according to actual needs. In addition, existing language-prompt-based anomaly detection methods (e.g., WinCLIP) also face similar issues, as they require offline computation of normal or anomaly textual features for all objects or textures.
>
>
> + **sm3a-Q5**: It is unclear whether a single example can represent all normal examples for a class effectively. Searching for the best matching normal image for each query image is resource-intensive.
>
> As shown in Table 1, our method requires only one normal image prompt to achieve competitive performance when faced with an open-world anomaly detection scenario. This indirectly supports that one normal image can essentially represent the normal pattern for a specific category. Undoubtedly, increasing the number of normal image prompts can further enhance performance, but this involves additional computational costs.
>
> In our MetaUAS*, we have demonstrated that the optimal normal prompt yields better performance compared to a random image prompt, although the matching process does indeed bring computational cost. Exploring other efficient algorithms to obtain a better prompt is also worthwhile, such as using image-matching to pre-align the prompt image with the query image.
>
>
> + **sm3a-Q6**: There are a few typos throughout the text, and the figures and tables could be placed differently.
>
> Thank you very much for pointing out the few typos. We will carefully review our manuscript and correct these typos in the revised version.

---

> > ### Comment · Reviewer_sm3a · 2024-08-12
> >
> > Thanks for the efforts in the rebuttal. I will maintain my score.

---

### Official Review · Reviewer_gb5F · 2024-07-12

**Soundness:** 3
**Presentation:** 3
**Contribution:** 3
**Rating:** 6
**Confidence:** 4

**Summary:**

The proposed method reformulates the one-shot anomaly detection task as a change detection task. The proposed method is trained with a synthesitzed change dataset, where objects are added, deleted or exchanged. Additionally, local changes are generated by pasting out-of-distribution textures on images in random locations. The proposed method is trained in a meta-learning manner where each meta-task is a prompt+query pair where a specific change must be detected and accurately segmented. The trained model is applied to industrial anomaly detection, where a single anomaly-free image is used as the prompt while a potentially anomalous image is used as the query. The model must then segment potential changes between the prompt and query image.

**Strengths:**

- The proposed method is interesting and it deviates significantly from the top performing few-shot anomaly detection methods since most recent methods are based on CLIP.
- The ablation study is mostly well done and individual components seem to be evaluated properly.
- The proposed method achieves good results
- The change detection dataset generation is an interesting approach to creating a dataset and it is interesting that the method generalizes well to industrial inspection datasets even though the training dataset is from a different domain.

**Weaknesses:**

- Clarity issues in the paper (Exactly how prompt pooling is performed in Section 3.4.)
- The way CLIP is added to the architecture in MetaAS*+ is not clear.
- Figure 4 - First column and the rest of the figure should be separated better. Currently it seems like the entire row depicts GT masks or queries.
- The evaluation of backbones is limited to convolutional models. The only transformer is used with the MetaAS*+ model.
- MetaAS*+ uses CLIP and achieves the best performance. Why this helps with the one-shot anomaly detection task is not intuitive and a further discussion would be beneficial for the reader.

**Questions:**

- Why are only convolutional networks used? Transformer networks should also work well here? A Transformer backbone in the ablation in Table 3b would be nice.
- How is the visual prior from CLIP introduced to the framework. In Table 2 the complexity of MetaAS*+ suggests that Eb4 and ViT are both run but this is not well explained.
- Couldn’t this be extended to the few-shot setup at inference by just adding more prompts and aligning features? If no, why not? If yes, why limit the evaluation to one-shot?

**Limitations:**

The authors have adequately addressed the limitations.

---

> ### Author Rebuttal · Authors · 2024-08-05
>
> + **gb5F-Q1**: A Transformer backbone in the ablation in Table 3b would be nice.
>
> Thanks for your suggestion. As pointed out by DDkp and sm3a, our method is compatible with various pre-trained models, including Convolutional and Transformer architecture networks. Considering the efficiency, we employ EfficientNet-b4 (E-b4) as our encoder, which is the same as in the previous UniAD.
>
> Following your suggestion, we replace the convolution-based EfficientNet-b4 with the recent EfficientViT [1]. Specifically, three EfficientViTs with different capacities (b1, b2 and b3) are used as our encoders, and the corresponding anomaly detection results on MVTec are reported in the following table. We can see that their performance is still lower than that of EfficientNet-b4 in most metrics. Other more powerful and efficient backbones deserve further exploration. In the revised version, we will add these results.
>
> |Backbone	|Total (M)	| Learnable (M) |I-ROC |I-PR |P-ROC |P-PR |P-PRO|
> | :---  |    :----:   |  :----:  |  :----: |  :----: |  :----: |  :----: |  :----:  |
> |E-b4 	|22.1	|4.6	|**91.3** |**96.2** |94.6 |**59.6** |**82.6** |
> |EVit-b1 |8.0	|3.4	|89.1 |94.7 |94.1 |55.8 |80.6 |
> |EVit-b2 |19.5	|4.5	|88.5 |94.9 |93.0 |56.3 |75.1 |
> |EVit-b3 |44.7	|5.7	|89.5 |95.7 |**95.3** |58.5 |80.9 |
>
>
>
> + **gb5F-Q2**: How the CLIP's visual prior is introduced into the framework.
>
> We apologize for the confusion about MetaUAS*+. In fact, we have given more details of diverse state-of-the-art methods, our MetaUAS and its variants (MetaUAS* and MetaUAS*+) in the Sec. B of Supplementary Materials. Here, we give some explanations about MetaUAS, MetaUAS\* and MetaUAS*+, and hope to address your concerns.
>
> The MetaUAS is capable of segmenting any anomalies only just by giving one normal image prompt. Note that the normal image prompt is randomly selected from a specific normal training set in MetaUAS. Different from MetaUAS, the MetaUAS* searches the best-matched normal image as a vision prompt from the normal training set for each query image. The MetaUAS*+ builds on MetaUAS*, but introduces the visual prior from a CLIP model. The visual prior knowledge is obtained by computing the cosine similarity between the query feature and the corresponding prompt feature extracted from the CLIP vision encoder, which is the same as WinCLIP+ [2] and PromptAD [3].
>
> + **gb5F-Q3**: Whether the evaluation can be extended from one-shot to few-shot?
>
> Our method can be flexibly extended to a few-shot setting, where a few normal image prompts are provided. However, we seek to push the limits of simple but effective methods for more general and challenging settings of anomaly segmentation. To this end, we explore universal anomaly segmentation from the perspective of a framework paradigm, where synthesized images are considered for one-prompt meta-learning, geometrical variations between prompt and query images can be effectively handled, and the inference is more efficient and simple without any guidance from language or fine-tuning on target datasets.
>
> Here, we present a straightforward manner to extend our approach from one-shot to few-shot. We use an average of all shots’ predictions as the final anomaly segmentation map. In the following table, we can observe a significant performance improvement from 1-shot to 3-shot. However, the performance tends to stabilize when reaching 5-shot. We leave it to future work on how to more elegantly extend MetaUAS from one-shot to few-shot.
>
> |Shot	    |I-ROC |I-PR |P-ROC |P-PR |P-PRO|
> | :---  |    :----:   |  :----:  |  :----: |  :----: |  :----: |
> |1			|91.3 |96.2 |94.6 |59.6 |82.6|
> |3			|92.7 |96.9 |95.6 |63.4 |85.5|
> |5			|93.0 |97.1 |95.9 |63.9 |86.1|
>
> + **gb5F-Q4**: How prompt pooling is performed?
>
> We obtain a feature representation by global average pooling on the highest-level feature from the encoder. For a class-agnostic query image, we match the corresponding normal image prompt using the cosine similarity between the query feature representation and all offline prompt feature representations.  The above details have been given in Lines 265-270.
>
>
> + **gb5F-Q6**: A further discussion of MetaUAS*+.
>
> Thanks for your suggestion. The MetaUAS*+ can be seen as an ensemble of MetaUAS* and the visual prior knowledge of CLIP. Therefore, the performance improvement is intuitive from an ensemble learning perspective. We will add the discussion to the revised version.
>
>
> [1] EfficientViT: Lightweight Multi-Scale Attention for High-Resolution Dense Prediction, ICCV 2023.
>
> [2] WinCLIP: Zero-/Few-Shot Anomaly Classification and Segmentation, CVPR 2023.
>
> [3] PromptAD: Learning Prompts with only Normal Samples for Few-Shot Anomaly Detection, CVPR 2024.

---

> > ### Comment · Reviewer_gb5F · 2024-08-13
> >
> > Thanks for the insightful rebuttal. The adaptation to the few-shot scenario is interesting. It would also be interesting to see the performance of the MetaUAS* on the few-shot setup due to the difference in performance to MetaUAS in the one-shot scenario. But otherwise my concerns have been addressed.

---

> > > ### Author Response · Authors · 2024-08-14
> > > **Response to Reviewer Concerns on Few-Shot MetaUAS***
> > >
> > > We are pleased to hear that our rebuttal has addressed most of your concerns, and we appreciate your continued engagement in this discussion. In response to your interest in the few-shot MetaUAS*, we have included the results in the table below.
> > >
> > > |Shot |I-ROC |I-PR |P-ROC |P-PR |P-PRO|
> > > | :---: | :---: | :---: | :---: | :---: | :---: |
> > > |1	 |94.2 |97.6 |95.3  |63.7  |83.1|
> > > |3      |95.0 |98.0 |96.2  |66.0  |85.9|
> > > |5	 |**95.2** |**98.1** |**96.4**  |**66.4**  |**86.4**|
> > >
> > > It is important to note that the one-shot MetaUAS* is based on the best-matched normal image prompt. Consequently, for the 3-shot and 5-shot MetaUAS*, we utilized the top 3 and top 5 normal image prompts, respectively. As shown, the performance trend of MetaUAS* is similar to that of MetaUAS (gb5F-Q3) as the shot number of normal image prompts increases.
> > >
> > > We acknowledge that our current extension from one-shot to few-shot is quite straightforward. We consider it a preliminary approach, leaving room for future work.
> > >
> > > Thank you for your feedback.

---

### Official Review · Reviewer_DDkp · 2024-07-12

**Soundness:** 3
**Presentation:** 4
**Contribution:** 4
**Rating:** 7
**Confidence:** 5

**Summary:**

This paper considers the anomaly segmentation task as a change segmentation task.  Then the large-scale image pairs with object-level and local region changes are synthesized to train a universal anomaly segmentation framework MetaUAS.  This only needs one normal image as the prompt.  The soft feature alignment module is proposed to handle geometrical variations between
prompt and query images.  This method achieves state-of-the-art performance on MVTec AD, VisA, and Goods datasets.

**Strengths:**

1. This paper synthesizes large-scale image pairs with object-level and local region changes, and desiges a universal anomaly segmentation framework that only need one normal image as prompt. This idea is novel and has practical implications.
2. This paper propose the soft feature alignment module to handle geometrical variations between prompt and query images. This is a novel, simple and effective alignment method.
3. This method is compatible with various feature extractors, and is not limited to CLIP like some previous methods.
4. The authors conducted experiments on three anomaly detection datasets. Significant improvement has been achieved, which proves the powerful generalization ability of this method.

**Weaknesses:**

1. The three fused features and two low-level original features are input into decoder. How these features are used in the decoder needs to be explained in more detail.
2. CLIPSeg [1] also uses one image as the prompt to perform segmentation tasks. It is beneficial to discuss and compare with CLIPSeg in this paper.

[1] Lüddecke T, Ecker A. Image segmentation using text and image prompts[C]//Proceedings of the IEEE/CVF conference on computer vision and pattern recognition. 2022: 7086-7096.

**Questions:**

See the weaknesses.

**Limitations:**

None.

---

> ### Author Rebuttal · Authors · 2024-08-05
>
> + **DDkp-Q1**: How these features are used in the decoder needs to be explained in more detail.
>
> Our method implements the decoder using standard UNet. Furthermore, we also compare the UNet with the FPN decoder in Table 3(c). In our paper, we omit the details of UNet and FPN because they are two popular modules that have been standardized and integrated into the [PyTorch library](https://github.com/qubvel-org/segmentation_models.pytorch/tree/main/segmentation_models_pytorch).  As we know, the decoder of UNet and FPN is widely used in dense visual prediction tasks, such as semantic segmentation and object detection. They use a top-down structure with lateral connections to fuse multi-scale input features and output a single high-resolution feature map. Considering reproducibility, we will make the codes and models of MetaUAS available.
>
>
> + **DDkp-Q2**: It is beneficial to discuss and compare with CLIPSeg.
>
> Thank you for pointing out a related work, CLIPSeg [1]. The CLIPSeg has one-shot semantic segmentation capabilities by providing one support image to the pre-trained CLIP model. However, its better performance still struggles with the powerful CLIP model and textual prompts. Different from the CLIPSeg, our method is compatible with various pre-trained models and does not require any guidance from language, as pointed out by you and Reviewer sm3a.
>
> [1] Image Segmentation Using Text and Image Prompts, CVPR 2022.

---

> > ### Comment · Reviewer_DDkp · 2024-08-08
> > **Response to Author Rebuttal**
> >
> > Thank the authors for their response. I'm inclined to keep my rating. This is a very interesting and novel paper, but this method does not seem to be very robust to different normal image prompts, which makes it impossible to continue to improve the rating to Strong Accept.

---

> > > ### Author Response · Authors · 2024-08-09
> > > **Response to Reviewer Concerns on Robustness**
> > >
> > > Thank you for your thoughtful feedback and for maintaining your rating (**Accept**). We appreciate your recognition of the novelty and interest in our work. We understand your concerns regarding the robustness of our method to different normal image prompts and would like to address these as follows to clarify our approach further.
> > >
> > > Firstly, as demonstrated in Figure 4, our MetaUAS shows robustness to different normal image prompts within the same category, particularly in those categories with significant geometric variations, such as grid and screw. This indicates that our method can effectively handle different prompts within the same category.
> > >
> > > Secondly, to quantitatively evaluate the robustness of our MetaUAS, we report performance metrics including the mean and variance based on the results from five different normal image prompts (generated with random seeds). It can be observed that our MetaUAS achieves higher means and lower variances compared to WinCLIP+ across most metrics, demonstrating its superior stability.
> > >
> > > Lastly, we have found that matching the optimal prompt (as shown in MetaUAS* in Table 1) or using few-shot normal prompts (e.g., 5-shot) can further result in significant performance improvements or enhance robustness (as detailed in the gb5F-Q3).
> > >
> > > We hope these explanations adequately address your concerns regarding the robustness of our method. We are grateful for the opportunity to discuss these aspects further and thank you once again for your valuable feedback.

---

### Author Rebuttal · Authors · 2024-08-05

We thank all reviewers (DDkp, gb5F, sm3a and bbUz) for your insightful comments. The reviewers believe that the proposed universal anomaly segmentation framework is **novel and interesting** (DDkp, gb5F and sm3a), **simple, effective and efficient** (DDkp and sm3a) and **compatible** (DDkp), the ablation study is **well** (gb5F and sm3a), our method achieves **good and encouraging results, or powerful generalization ability** (DDkp, gb5F, sm3a and bbUz).

Next, we respond to the concerns of each reviewer one by one.

---

### Decision · Program_Chairs · 2024-09-25

**Decision:**

Accept (poster)

**Comment:**

This paper proposes to frame anomaly segmentation as a change segmentation task. It proposes a meta-learning approach, trained on a change segentation dataset where image pair changes are artificially created. All Reviewers provided positive feedback for this paper, praising the framing of the problem and the results achieved by the proposed method. Given the unanymous positive feedback received by this paper, I recommend acceptance.